# CQE: A Comprehensive Quantity Extractor

**Satya Almasian,**[*] **Vivian Kazakova**[*]**, Philipp Göldner** and **Michael Gertz**
Institute of Computer Science, Heidelberg University, Germany
`{almasian,gertz}@informatik.uni-heidelberg.de`
`{vivian.kazakova,goeldner}@stud.uni-heidelberg.de`

## Abstract

Quantities are essential in documents to describe factual information. They are ubiquitous in application domains such as finance, business, medicine, and science in general. Compared to other information extraction approaches, interestingly only a few works exist that describe methods for a proper extraction and representation of quantities in text.

In this paper, we present such a comprehensive quantity extraction framework from text data. It efficiently detects combinations of *values* and *units*, the behavior of a quantity (e.g., rising or falling), and the *concept* a quantity is associated with. Our framework makes use of dependency parsing and a dictionary of units, and it provides for a proper normalization and standardization of detected quantities. Using a novel dataset for evaluation, we show that our open source framework outperforms other systems and – to the best of our knowledge – is the first to detect concepts associated with identified quantities. The code and data underlying our framework are available at `https://github.com/vivkaz/CQE`.

## 1 Introduction

Quantities are the main tool for conveying factual and accurate information. News articles are filled with social and financial trends, and technical documents use measurable values to report their findings. Despite their significance, a comprehensive system for quantity extraction and an evaluation framework to compare the performance of such systems are not yet at hand. In the literature, a few works directly study quantity extraction, but their focus is limited to physical and science domains ([Foppiano et al., 2019](#)). Quantity extraction is often part of a larger system, where identification of quantities is required to improve numerical understanding in retrieval or textual entailment tasks ([Roy et al., 2015](#); [Li et al., 2021](#); [Sarawagi](#)

and [Chakrabarti, 2014](#); [Banerjee et al., 2009](#); [Maiya et al., 2015](#)). Consequently, their performance is measured based on the downstream task, and the quality of the extractor, despite its contribution to the final result, is not separately evaluated. Therefore, when in need of a quantity extractor, one has to resort to a number of open source packages, without a benchmark or a performance guarantee. Since quantity extraction is rarely the main objective, the capabilities of the available systems and their definition of quantity vary based on the downstream task. As a result, the context information about a quantity is reduced to the essentials of each system. Most systems consider a quantity as a number with a measurable and metric *unit* ([Foppiano et al., 2019](#)). However, outside of scientific domains any noun phrase describing a *value* is a potential *unit*, e.g., "5 bananas". Moreover, a more meaningful representation of quantities should include their behaviour and associated *concepts*. For example, in the sentence "DAX fell 2% and S&P gained more than 2%", the *value/unit* pair ⟨*2, percentage*⟩ indicates two different quantities in association with different concepts, DAX and S&P, with opposite behaviours, *decreasing* and *increasing*. These subtleties are not captured by simplified models.

In this paper, we present a comprehensive quantity extraction (CQE) framework. Our system is capable of extracting standardized *values*, physical and non-physical *units*, *changes* or trends in values, and *concepts* associated with detected values. Furthermore, we introduce NewsQuant, a new benchmark dataset for quantity extraction, carefully selected from a diverse set of news articles in the categories of economics, sports, technology, cars, science, and companies. Our system outperforms other libraries and extends on their capabilities to extract *concepts* associated with values. Our software and data are publicly available. By introducing a strong baseline and novel dataset, we aim to motivate further research and development in this field.

---

[*]These authors contributed equally to this work.

## 2 Related Work

In literature, quantity extraction is mainly a component of a larger system for textual entailment or search. The only work that solely focuses on quantity extraction is Grobid-quantities (Foppiano et al., 2019), which uses three Conditional Random Field models in a cascade to find *value/unit* pairs and to determine their relation, where the *units* are limited to the scientific domain, a.k.a. *SI units*.

(Roy et al., 2015)'s definition of a quantity is closer to ours and is based on Forbus' theory (Forbus, 1984). A quantity is a (*value*, *unit*, *change*) triplet, and noun-based units are also considered. Extraction is performed as a step in their pipeline for quantity reasoning in terms of textual entailment. Although they only evaluate on textual entailment, the extractor is released as part of the CogComp natural language processing libraries, under the name Illinois Quantifier.[1]

Two prominent open source libraries for quantity extraction are (a) Recognizers-Text (Huang et al., 2017; Chen et al., 2023) from Microsoft and (b) Quantulum3.[2] Recognizers-Text uses regular expressions for the resolution of numerical and temporal entities in ten languages. The system has separate models for the extraction of *value/unit* pairs for percentages, age, currencies, dimensions, and temperatures and is limited to only these quantity types. Moreover, it cannot proactively distinguish the type of quantity for extraction and the user has to manually select the correct model.

Quantulum3 uses regular expression to extract quantities and a dictionary of *units* for normalization. For *units* with similar surface forms, a classifier based on Glove embeddings (Pennington et al., 2014) is used for disambiguation, e.g., "pound" as weight or currency.

Recognizers-Text is used in the work of (Li et al., 2021) to demonstrate quantity search, where the results are visualized in the form of tables or charts. They define quantity facts as triplets of (*related, value & unit, time*). *Related* is the quantitative related information, close to our definition of *concept*. However, it is not part of their quantity model but rather extracted separately using rules. They utilize the quantity facts for the visualization of results but do not evaluate their system or the quantity extrac-

tion module. QFinder (Almasian et al., 2022) uses Quantulum3 in a similar way to demonstrate quantity search on news articles, but does not comment on the extractor's performance.

Another system that indirectly considers *concepts* is Xart (Berrahou et al., 2017), where instances of n-ary relations containing numerical values and unit are extracted and concepts are an argument in these relations. However, the concepts are limited to a domain ontology with specific concepts of a given application domain.

A number of other works utilize quantity extraction as part of their system. MQSearch (Maiya et al., 2015) extracts quantities with a set of regular expressions for a search engine on numerical information. Qsearch (Ho et al., 2019) is another quantity search system, based on quantity facts extracted with the Illinois Quantifier. The works by (Banerjee et al., 2009; Sarawagi and Chakrabarti, 2014) focus on scoring quantity intervals in census data and tables.

## 3 Extraction of Quantities

In the following, we describe our quantity representation model and detail our extraction technique.

### 3.1 Quantity Representation

In general, anything that has a count or is measurable is considered a quantity. We extend upon the definition by (Roy et al., 2015) to include concepts and represent a quantity by a tuple $\langle v, u, ch, cn \rangle$ with the following components:

1. *Value (v):* A real number or a range of values, describing a magnitude, multitude, or duration, e.g., "the car accelerates from 0 to 72 km/h", has a range of $v = (0, 72)$ and, "the car accelerated to 72 km/h" has a single value $v = 72$. *Values* come in different magnitudes, often denoted by prefixes, and sometimes containing fractions, e.g., "He earns 10k euros" $\rightarrow v = 10000$, or "1/5 th of his earnings"$\rightarrow v = 0.2$.

2. *Unit (u):* A noun phrase defining the atomic unit of measure. *Units* are either part of a predefined set of known scientific and monetary types, or in a more general case, are noun phrases that refer to the multitude of an object, e.g., "2 apples" $\rightarrow u = apple$ (Rijgersberg et al., 2013). The predefined set corresponds either to (a) *scientific units* for measurement of physical attributes (e.g., "2km" has the *scientific unit* ($u = kilometre$)),

[1]https://github.com/CogComp/cogcomp-nlp/tree/master/quantifier Last accessed: October 16, 2023

[2]https://github.com/nielstron/quantulum3 Last accessed: October 16, 2023

or (b) *currencies*, as the unit of money (e.g., "10k euros" refers to a currency). Predefined *units* can have many textual or symbolic surface forms, e.g., "euro", "EUR", or "€", and their normalization is a daunting task. Sometimes the surface forms coincide with other units, resulting in ambiguity that can only be resolved by knowing the context, e.g., "She weighs 50 pounds" is a measure of weight ($u = $ pound-mass) and not a currency.

3. *Change (ch):* The modifier of the quantity *value*, describing how the *value* is changing, e.g., "roughly 35$" is describing an approximation. (Roy et al., 2015) introduce four categories for *change*: $=$ (equal), $\sim$ (approximate), $>$ (more than), and $<$ (less than). These categories are mainly describing the bounds for a quantity. We extend this definition by accounting for trends and add two more categories: $up$ and $down$ for increasing and decreasing trends, e.g., "DAX fell 2%" indicates a downward trend ($ch = down$), while "He weighs more than 50kg" is indicating a bound ($ch = $ '$>$').

4. *Concept (cn): Concepts* are subjects describing or relating to a value. A quantity mentioned in a text is either measuring a property of a phenomenon, e.g., "height of the Eiffel Tower", in which case the phenomenon and the property are the concepts, or an action has been made, involving a quantity, e.g., "Google hired 100 people", in which case the actor is what the quantity is referring to. In the phrase "DAX fell 2%" the quantity is measuring the worth of $cn = DAX$ or in "The BMW Group is investing a total of $200 million" the investment is being made by $cn = BMW\ Group$. Sometimes a *concept* is distributed in different parts of a sentence, e.g., "The iPhone 11 has 64GB of storage. " $\rightarrow cn = iPhone\ 11, storage$. A *concept* may or may not be present, e.g., "200 people were at the concert" has no concept.

## 3.2 Quantity Extraction

Similar to previous work, we observed that quantities often follow a recurring pattern. But instead of relying on regular expressions, we take advantage of linguistic properties and dependency parsing. The input of our system is a sentence, and the output is a list of detected quantities.
**Example 1:** "In Europe, German DAX fell 0.4 pc, while the CAC40 in France gained 0.1." results in

- $\langle v = 0.4,\ u = percentage,$
  $ch = down,\ cn = (German, DAX)\rangle$

- $\langle v = 0.1,\ u = percentage,$
  $ch = up,\ cn = (CAC40, France)\rangle$.

### 3.2.1 Pre-processing
The pre-processing stage includes the removal of unnecessary punctuations, e.g., "m.p.h" $\rightarrow$ "mph", the addition of helper tokens, and other text cleaning steps. An example of a helper token is placing a *minus* in front of negative values for easy detection in other steps. These steps are done prior to dependency parsing and POS tagging to improve their performance. Numerals that do not fit the definition of a quantity, such as phone numbers and dates, are detected with regular expressions and disregarded in further steps.

### 3.2.2 Tokenization
We perform a custom task-specific word tokenization. Our tokenizer is aware of separator patterns in *values* and *units* and avoids between-word splitting. For example, in the sentence "A beetle goes from 0 to 80 km/h in 8 seconds.", a normal tokenizer would split $km/h \rightarrow (km, /, h)$ but we will keep the *unit* token intact. Another example is a numerical token containing punctuations, e.g., $2.33E$-3, where naive tokenization changes the value.

### 3.2.3 Value, Unit, and Change Detection
The tokenized text is matched against a set of rules based on a dependency parsing tree and POS tags. A set of 61 rules was created based on patterns observed in financial data and scientific documents and by studying previous work (Maiya et al., 2015; Huang et al., 2017). A comprehensive list of all rules can be found in the repository of our project. The rules are designed to find tokens associated with *value*, *unit*, and *change*.
*Value/unit* pairs are often sets of numbers and nouns, numbers and symbols, or number and adjectives in various sentence structures. For ranges, the rules become more complex, as lower and upper bounds need to be identified using relational keywords such as "from... to" or "between".
*Changes* are often adjectives or verbs that have a direct relation to a number and modify its value. Sometimes symbols before a number are also an indication of a *change*, e.g., "$\sim 10$" describes an approximation. In general, there are six *change* categories. $\sim$ for approximate equality, $=$ for exact equality, $>$ for greater than bounds, $<$ for less than

bounds, *up* denoting an increasing or upward trend, and *down* for decreasing or downward trend.

As an example of the extraction, we look at *value*, *unit* and *change* detection for the two quantities in Example 1. Note that in this stage the surface forms are detected and not normalized values, e.g., "pc" versus "percentage".

The NOUN_NUM rule detects the surface form for the first *value/unit* pair, (0.4, pc). Here, the *value* has NUM as a POS-tag and is the immediate syntactic dependent of the *unit* token, which is a noun or proper noun.

The LONELY_NUM rule detects the *value/unit* pair for the second quantity, namely *(0.1, None)*. If all other rules fail to find a *value/unit* pair, this rule detects the number with the POS-tag NUM.

QUANTMOD_DIRECT_NUM detects the *change*, by looking at the verb or adjective directly before NUM tokens. Here, "fell" is a trigger word for a downward trend. For Example 1, we thus have two extracted triplets with *value*, *unit*, and *change*.

- $\langle v = 0.4,\ u = pc,\ ch = fell \rangle$

- $\langle v = 0.1,\ u = None,\ ch = gained \rangle$,

More examples are given In Appendix A.1.

If no *unit* is detected for a quantity, its context is checked for the possibility of *shared units*. For the quantity $\langle v = 0.1,\ u = None,\ ch = gained \rangle$ in Example 1 ,"percentage" is the derived *unit*, although not mentioned in the text. *Shared units* often occur in similarly structured sub-clauses or after connector words such as "and", "while", or "whereas". The similarity between two sub-clauses is computed using the *Levenshtein ratio* between the structure of clauses. The structure is represented by POS-tags, e.g., "German DAX fell 0.4 pc" → "JJ NNP VBD CD NN" and "the CAC40 in France gained 0.1" →"DT NNP IN NNP VBD CD". This ratio is between 0 and 100, where larger values indicate higher similarity. If connector words are present and the ratio is larger than 60, the unitless quantity is assigned the *unit* of the other sub-clause, e.g., $None$ becomes $pc$.

Finally, the candidate *values* are filtered by logical rules to avoid false detection of *non-quantities*, e.g., in "S&P 500", 500 is not a quantity.

### 3.2.4 Concept Detection

*Concepts* are detected in one of the following five ways, ordered by priority:

1. Keywords, such as *for*, *of*, *at* or *by* before or after a *value* point to a potential concept. For example, "with carbon levels *at* 1200 parts per million" results in $cn = (carbon,\ levels)$. The noun and pronouns before and after such keywords are potential concepts.

2. The entire subtree of dependencies with a number (*value*) as one of the leaf nodes is inspected to find the closest verb related to the number. If no verb is found, then the verb connected to the ROOT is selected. The nominal subject of the verb is considered as the *concept*. In Example 1, both "German DAX" and "CAC40 in France" are the nominal subjects of the closest verbs to the *values* in the text.

3. Sometimes values occur in a relative clause that modifies the nominal, e.g., "maximum investment per person, which is 50000" → $cn = (maximum,\ investment,\ per,\ person)$. In such a case, the noun phrase before the relative clause is the *concept*, since the relative clause is describing it.

4. If the numerical *value* in a sentence is not associated with the nominal of the sentence, then it is mostly likely related to the object. Therefore, the direct object of the verb is also a candidate, e.g., "She gave me a raise of $1k", where "raise" is the direct object of the verb.

5. Finally, if the *concept* is not found in the previous steps, and there is a single noun in the sentence, the noun is tagged as the *concept*, e.g., "a beetle that can go from 0 to 80 km/h in about 8 seconds, " → $cn = (beetle)$.

From the list of candidate tokens for *concepts*, tokens previously associated with *units* and *values* are filtered and stopwords are removed, e.g., "CAC40 in France" results in $cn = (CAC40,\ France)$. Generally, a *concept* is represented as a list of tokens.

### 3.2.5 Normalization and Standardization

The final stage is the normalization of *units* and *changes* using dictionaries and standardization of *values*. The *units dictionary* is a set of 531 *units*, their surface forms and symbols gathered from the Quantulum3 library, a dictionary provided by Unified Code for Units of Measure (UCUM) (Lefrançois and Zimmermann, 2018), and

a list of *units* from Wikipedia.[3] An example of an entry in this dictionary for "euro" is:

```
{"euro":
"surfaces": ["Euro","Euros","euro",
"euros"],
"symbols": ["EUR","eur",€]}
```

The detected token span of a *unit* is normalized by matching against the different surface forms and symbols in the dictionary. The normalized form is the key of the dictionary and is added to the output, e.g., "euro" in the example above or "cm" giving "centimetre". The normalization makes the comparison of different units easier. Note that conversions between metric units is not supported. For example, "centimetre" is kept as the final representation and not converted to "metre".

If the detected surface form is shared across multiple *units*, the unit is *ambiguous* and requires further normalization based on the context. Since language models are great at capturing contextual information, for this purpose, we train a BERT-based classifier (Devlin et al., 2019). There are 18 ambiguous surface forms in our unit dictionary, and for each a separate classifier is trained that allows to distinguish among *units* based on the context. If an ambiguous surface form is detected by the system, the relevant classifier is used to find the correct normalized unit.

*Compound units* are also detected and normalized independently. For example, "kV/cm" results in "kilovolt per centimetre', where "kV" and "cm" are normalized based on separate dictionary entries.

If no valid match in the dictionary exists, the surface form is tagged as a *noun unit* and lemmatized, e.g., "10 students" gives $u = student$. In some cases, the adjective before a noun is also part of the unit, e.g., "two residential suites" results in $u = residential\ suite$.

The *value dictionary* contains the necessary information to standardize *values* to real numbers. More specifically, it contains surface forms for prefixes and suffixes of scales, e.g., "B: billion" or "n: nano", spelled out numbers in textual format, e.g., "fourty-two: 42", fractions in textual format, e.g., "half: 1/2", and scientific exponents, e.g., "$10^2$: 100'. This combination is used to convert *values* to decimal format. Scientific notations with exponent and mantissa are converted to decimal values, e.g.,"$2.3E2 \rightarrow v = 23$".

[3]https://en.wikipedia.org/wiki/Template: Convert/list_of_units Last accessed: April 17, 2023

Various trigger words or symbols for bounds and trends are managed in the *changes dictionary*, where detected tokens for *change* are mapped to one of the allowed categories $\sim, =, >, <, up, down$. For example, the entry for equality is "=": [ "exactly", "just", "equals", "totalling","="].

## 4 Evaluation

CQE is compared against *Illinois Quantifier* (IllQ), *Quantulum3* (Q3), *Recognizers-Text* (R-Txt), *Gorbid-quantities* (Grbd) and GPT-3 with few-shot learning (Brown et al., 2020). From here on, the abbreviations are used to refer to the respective system. We first compare the functionality of the models, then describe our benchmark dataset and compare the models on precision, recall and F1-score for quantity extraction. Finally, the unit disambiguation module is evaluated on a custom-made dataset against Q3. Our evaluation code and datasets are available at https://github.com/satya77/CQE_Evaluation.

### 4.1 Comparison of Functionality

Table 1 compares the functionality of the models in terms of different types of *values*, *units*, and *changes*, as well as normalization techniques.

IllQ is the only baseline that is able to detect *changes* in *values* but in a limited setting that does not consider upward or downward trends. IllQ performs normalization for currencies, however, *scientific units* are not normalized. Furthermore, it fails to detect *fractional values* and *ranges*.

After our approach (CQE), Q3 has the most functionality and is the only model that correctly detects *ranges* and *shared units* and performs *unit* disambiguation. On the other hand, Q3 disregards *noun-based units*, and although it is capable of detecting a wide range of *value* types, it makes incorrect detections of *non-quantitative values*.

R-Txt has dedicated models for certain quantity types but fails to detect other types in the text, ignoring *ranges*, *scientific notation*, and *noun-based units*. The unit normalization is limited to the quantity types and lacks disambiguation.

Grbd model's major shortcoming is the lack of value standardization, where fractions such as "1/3" and scaled values like "2 billion" are not standardized correctly. The system is limited to scientific units, and unit normalization works differently than another system, where the scientific units are con-

Table 1: Comparison of functionality for various extractors.

| Feature | Example | CQE | IllQ | R-Txt | Q3 | Grbd |
|---|---|---|---|---|---|---|
| Value | 5k euros (5k) | ✓ | ✓ | ✓ | ✓ | ✓ |
| Standardization | 5k euros (5000) | ✓ | ✓ | ✓ | ✓ | ✗ |
| Negative Values | -5 C (-5) | ✓ | ✗ | ✓ | ✓ | ✓ |
| Fractions | 1/3 of the population (0.33) | ✓ | ✗ | ✓ | ✓ | ✓ |
| Range | 40-60 km/h (40-60) | ✓ | ✗ | ✗ | ✓ | ✓ |
| Non-quantities | iPhone 11 (-) | ✓ | ✗ | ✗ | ✗ | ✓ |
| Scientific Notation | $1.9 \times 10^2$ (190) | ✓ | ✗ | ✗ | ✓ | ✗ |
| Unit | 1mm (mm) | ✓ | ✓ | ✓ | ✓ | ✓ |
| Unit normalization | 1mm (millimetre) | ✓ | ✗ | ✓ | ✓ | ✗ |
| Unit disambiguation | 10 pound (sterling or mass?) | ✓ | ✗ | ✗ | ✓ | ✗ |
| Noun Units | 200 people (people) | ✓ | ✓ | ✗ | ✗ | ✓ |
| Shared Units | about 8 or $9 (both dollar) | ✓ | ✗ | ✗ | ✓ | ✓ |
| Change | more than 100 (>) | ✓ | ✓ | ✗ | ✗ | ✗ |
| Trends | DAX fell 2% (down) | ✓ | ✗ | ✗ | ✗ | ✗ |
| Concept | AAPL rose 2% (AAPL) | ✓ | ✗ | ✗ | ✗ | ✗ |

Table 2: Statistics of the number of sentences, quantities, and sentences with and without quantities in the NewsQuant and R-Txt datasets.

| Dataset | #sent | #quantity | #sent with quantity | #sent w/o quantity |
|---|---|---|---|---|
| NewsQuant | 590 | 904 | 475 | 115 |
| R-Txt-currencies | 180 | 255 | 178 | 2 |
| R-Txt-dimension | 93 | 121 | 77 | 14 |
| R-Txt-temperature | 36 | 34 | 34 | 2 |
| R-Txt-age | 19 | 22 | 18 | 1 |

verted to the base unit, and values are also scaled accordingly. For example, "1mm" is converted to (0,001, metre). GPT-3 has a lot of variability in the output and does not provide concrete and stable functionality like the models discussed in this section. Therefore, it is not further considered in this comparison.

## 4.2 NewsQuant Dataset

For a qualitative comparison, we introduce a new evaluation resource called NewsQuant, consisting of 590 sentences from news articles in the domains of economics, sports, technology, cars, science, and companies. To the best of our knowledge, this is the first comprehensive evaluation set introduced for quantity extraction. Each sentence is tagged with one or more quantities containing *value*, *unit*, *change*, and *concept* and is annotated by the two first authors of the paper. Inter-annotator agreements are computed separately for *value*, *unit*, *change*, and *concept* between the two first authors on a subset of 20 samples. For the first three, the Cohen Kappa coefficient (Cohen, 1960) with values of 1.0, 0.92, and 0.85 is reported. Value detection is a simpler task for humans and annotators

have perfect agreement. A *concept* is a span of tokens in the text and does not have a standardized representation, therefore, Cohen Kappa coefficient cannot be used. Instead, we report Krippendorff's alpha (Krippendorff, 2004), with the value of 0.79. In total, the annotators completely agreed on all elements for 62% of the annotations.

We additionally evaluate four datasets available in the repository of R-Txt for age, dimension, temperature, and currencies[4]. These datasets contain only *unit/value* pairs. The original datasets only contained tags for a certain quantity type and would ignore other types, giving the R-Txt model an advantage. For example, in the R-Txt-currencies, only the currencies were annotated, and other quantities were ignored. We added extra annotations for all other types of quantities for a fair comparison. For example, in the sentence "I want to earn $10000 in 3 years" from the currency dataset, where only "$10000" was annotated, we add "3 years". Statistics of the number of sentences and quantities for each dataset are shown in Table 2. The NewsQuant

---

[4]https://github.com/microsoft/Recognizers-Text/tree/master/Specs/NumberWithUnit/English Last accessed: October 16, 2023

dataset is the largest dataset for this task containing over 900 quantities of various types. NewsQuant is designed to test for the functionalities mentioned in Table 1 and includes negative examples with non-quantity numerals.

### 4.3 Disambiguation Dataset

To train our unit disambiguation system, a dataset of 18 ambiguous surface forms is created using ChatGPT[5]. For each ambiguous surface form, at least 100 examples are generated, and the final training dataset consists of 1,835 sentences with various context information. For more challenging surface forms, more samples are generated. For the list of ambiguous surface forms and the number of samples for each class, refer to Appendix A.3. A test dataset is generated in the same manner using ChatGPT, consisting of 180 samples, 10 samples per surface form. For more information on the dataset creation, please see Appendix A.4.

### 4.4 Implementation

CQE is implemented in Python 3.10. For dependency parsing, part-of-speech tagging, and the matching of rules SpaCy 3.0.9[6] is used. The unit disambiguation module, with BERT-based classifiers, is trained using spacy-transformers[7] for a smooth intergeneration with other SpaCy modules. Parsers were created to align the output format of different baselines so that the differences in output representation do not affect the evaluation. For instance, for IllQ, we normalize the *scientific units* and account for differences in the representation of ranges in Q3. If a value is detected by a baseline but not standardized or a unit is not normalized to the form present in the dataset, post-processing is applied for a unified output. These steps do not hurt the performance of the baseline models but rather align their output to the format of the benchmark dataset. For more details refer to Appendix A.2.

Moreover, to keep up with the recent trends in NLP and the lack of a baseline for *concept* detection, we introduce a GPT-3 baseline. The GPT-3 model is prompted to tag quantities with 10 examples for few-shot learning. Prompts and examples are available in our repository. We use the *text-*

*davinci-003* model from the GPT-3 API[8] with a sequence length of 512, temperature of 0.5, and no frequency or presence penalty. For more details, refer to Appendix A.2. We are aware that with extensive fine-tuning and more training examples GPT-3 values are likely to improve. However, the purpose of this paper is neither prompt engineering nor designing training data for GPT-3, and the few-short learning should suffice for a baseline.

### 4.5 Analysis of Results

All the models are compared on precision, recall, and F1-score for the detection of *value*, *unit*, *change*, and *concept*. Disambiguation systems are also compared regarding precision, recall, and F1-score of unit classification. Permutation resampling is used to test for significant improvements in F1-scores (Riezler and Maxwell, 2005), which is statistically more coherent in comparison to the commonly paired bootstrap sampling (Koehn, 2004). Results denoted with † mark highly significant improvements over the best-performing baseline with a $p$-value $< 0.01$.

#### 4.5.1 NewsQuant:

Table 3 shows the result on the NewsQuant dataset. Since Q3, Grbd, and R-Txt do not detect changes, respective entries are left empty. CQE beats all baselines in each category by a significant margin, where most of the errors are due to incorrect extraction of the dependency parsing tree and part-of-speech tagging.

The second best model, Q3, scores highly for *value* detection, but misses all the noun base *units* and tends to overgeneralize tokens to *units* where none exist, e.g., in "0.1 percent at 5884", Q3, detects "at" as *percent per ampere-turn*. Q3 makes mistakes on different currencies and their normalization. We attribute this to their incomplete unit dictionary.

R-Txt works well for the quantity types with dedicated models, but all the other quantities are ignored or misclassified. One has to manually select a quantity type for the R-Txt, therefore, we ran all the available model types on each sentence, where any detected quantity is forced into the available model types, resulting in miss-classifications.

IllQ has trouble with *compound units*, e.g., "$2.1 per gallon" and tends to tag the word after a *value* as a unit, e.g., in "women aged 25 to 54 grew by 1%", *grew by* is the detected *unit*. Although IllQ is

---

[5] https://chat.openai.com/ Last accessed: October 16, 2023

[6] https://spacy.io/ Last accessed: October 16, 2023

[7] https://spacy.io/universe/project/spacy-transformers Last accessed: October 16, 2023

[8] https://platform.openai.com/ Last accessed: October 16, 2023

Table 3: Precision, recall, and F1-score for detection of *value*, *unit* and *change* on NewsQuant.

| Model | Value | | | Value+Unit | | | Value+Change | | |
|---|---|---|---|---|---|---|---|---|---|
| | P | R | F1 | P | R | F1 | P | R | F1 |
| CQE | **92.0** | **91.9** | **92.0**† | **85.6** | **85.5** | **85.6**† | **88.2** | **88.1** | **88.1**† |
| Q3 | 65.0 | 83.3 | 73.0 | 42.1 | 53.9 | 47.2 | - | - | - |
| IllQ | 50.6 | 66.0 | 57.3 | 32.8 | 42.8 | 37.1 | 44.2 | 57.6 | 50.0 |
| R-Txt | 59.7 | 82.2 | 69.1 | 29.6 | 40.7 | 34.2 | - | - | - |
| Grbd | 58.8 | 53.1 | 55.8 | 37.4 | 33.7 | 35.5 | - | - | - |
| GPT-3 | 72.1 | 69.1 | 70.6 | 60.3 | 57.9 | 59.1 | 53.1 | 50.9 | 51.9 |

Table 4: Precision, recall and F1-score for detection of *value* and *unit* on R-Txt Datasets.

| Model | Detect | currency | | | dimension | | | temperature | | | age | | |
|---|---|---|---|---|---|---|---|---|---|---|---|---|---|
| | | P | R | F1 | P | R | F1 | P | R | F1 | P | R | F1 |
| CQE | | **82.6** | 85.9 | **84.2** | 85.5 | 87.6 | 86.5 | 94.3 | 97.1 | 95.7 | **91.3** | 95.5 | 93.3 |
| Q3 | | 69.2 | 84.7 | 76.2 | 76.9 | **93.4** | 84.3 | 91.7 | 97.1 | 94.3 | **91.3** | 95.5 | 93.3 |
| IllQ | Value | 65.5 | 70.6 | 67.9 | 65.3 | 77.7 | 70.9 | 88.9 | 94.1 | 91.4 | 65.4 | 77.3 | 70.8 |
| R-Txt | | 67.4 | **91.8** | 77.7 | 73.6 | 90.1 | 81.0 | 91.9 | **100.0** | **95.8** | 77.8 | 95.5 | 85.7 |
| Grbd | | 46.6 | 35.3 | 40.2 | 75.8 | 59.5 | 66.7 | 84.0 | 61.8 | 71.2 | 60.0 | 27.3 | 37.5 |
| GPT-3 | | 50.5 | 54.9 | 52.6 | 80.2 | 80.2 | 80.2 | 93.5 | 85.3 | 89.2 | 92.3 | 54.5 | 68.6 |
| CQE | | **78.1** | 81.2 | **79.6**† | 78.2 | 80.2 | 79.2 | 91.4 | 94.1 | 92.8 | **91.3** | 95.5 | 93.3 |
| Q3 | Value | 29.5 | 36.1 | 32.5 | 56.5 | 68.6 | 61.9 | 61.1 | 76.5 | 74.3 | 82.6 | 86.4 | 84.4 |
| IllQ | +Unit | 41.8 | 41.6 | 45.1 | 43.4 | 52.1 | 47.5 | 30.6 | 32.4 | 31.4 | 42.3 | 50.0 | 45.8 |
| R-Txt | | 46.7 | 63.5 | 53.8 | 44.6 | 54.5 | 49.1 | **91.9** | **100.0** | **95.8** | 70.4 | 86.4 | 77.6 |
| Grbd | | 24.9 | 18.8 | 21.4 | 44.2 | 34.7 | 38.9 | 32.0 | 23.5 | 27.1 | 40.0 | 18.2 | 25.0 |
| GPT-3 | | 40.8 | 44.3 | 42.5 | 65.3 | 65.3 | 65.3 | 45.2 | 41.2 | 43.1 | 92.3 | 54.5 | 68.6 |

Table 5: Relaxed and strict matching, precision, recall and F1-score for *concept* detection on the NewsQuant.

| Model | Relaxed Match | | | Strict Match | | |
|---|---|---|---|---|---|---|
| | P | R | F1 | P | R | F1 |
| CQE | **76.2** | **76.1** | **76.1**† | **57.0** | **57.0** | **57.0**† |
| GPT-3 | 55.9 | 53.7 | 54.8 | 26.3 | 25.2 | 25.7 |

Table 6: Weighted micro-average precision, recall and F1-score on the *unit* disambiguation dataset.

| Model | P | R | F1 |
|---|---|---|---|
| CQE | **89.9** | **89.4** | **88.1**† |
| Q3 | 57.33 | 57.78 | 54.46 |

supposed to normalize currencies, in practice the normalization is limited and often currency symbols are not normalized. Moreover, trends are ignored by IllQ, and the model is biased to predict equality (=) for most *changes*, and other changes are rare.

The Grdb model detects the correct surface form for values in most cases, however, due to unstable standardization many standardized values are incorrect. Unit normalization is limited to a small subset of units, where percentages and compound units are mainly ignored.

GPT-3 achieves a score close to Q3 for the detection of *units* and *values* and close to IllQ for *changes*. Nevertheless, due to extreme hallucination, extensive post-processing of the output is required for evaluation, e.g., many of the values extracted were not actual numbers and units were not normalized. Moreover, GPT-3 often confuses value suffixes with *units*, e.g., "billion" or "million" and, despite the normalization prompt, fails to normalize *units* and required manual normalization for most detections.

### 4.5.2 R-Txt Dataset:

Evaluation results on the four quantity types of the R-Txt dataset are shown in Table 4, where our model once again outperforms all baselines on *value+unit* detection for all categories except for temperature. Nevertheless, for temperature, the R-Txt improvement over CQE is not statistically significant. The small size of the age and temperature dataset results in inconsistent significance testing. The closeness of *value* detection between models is due to the structure of the dataset. Most

*value*s have the surface form of a decimal, and the diversity of types like ranges, fractions, and non-quantities is negligible. For more details on the error analysis and common mistakes of each model on NewsQuant and R-Txt, see Appendix A.6.

### 4.5.3 Concept Detection:

Finally, *concept* detection is evaluated on the NewsQuant dataset. Results are shown in Table 5. Following the approach of UzZaman et al. (UzZaman et al., 2013) for evaluation, strict and relaxed matches are compared. A strict match is an exact token match between the source and target, whereas a relaxed match is counted when there is an overlap between the systems and ground truth token spans. Based on the scores we observe that *concept* detection is harder in comparison to *value+unit* detection. Even GPT-3 struggles with accurate predictions. Our algorithm for *concept* detection is limited to common cases and does not take into account the full complexity of human language, leaving room for improvement in future work. Moreover, in many cases, the concept is implicit and hard to distinguish even for human annotators. In general, our approach is more recall-oriented, as we keep any potential candidate from the concept detection step in the final result set, trying to capture as many concepts as possible. Hence, there is a big gap between partial and complete matches. However, since the method is rule-based, rules can be adjusted to be restrictive and precision-focused.

### 4.5.4 Unit Disambiguation:

CQE is compared against Q3 (the only other systems with disambiguation capabilities) in Table 6. Since the normalization of *units* is not consistent in the GPT-3 model and requires manual normalization, GPT-3 is left out of this study. All 18 classifiers are evaluated within a single system. The results are averaged by weighting the score of each class label by the number of true instances when calculating the average. CQE significantly outperforms Q3 on all metrics, and it is easily expendable to new surface forms and *units* by adding a new classifier. Since the training data is generated using ChatGPT, a new classifier can be trained using our paradigm and data generation steps, as shown in Appendix A.4. For a detailed evaluation of each class, see Appendix A.5.

## 5 Conclusion and Ongoing Work

In this paper, we introduced CQE, a comprehensive quantity extractor for unstructured text. Our system is not only significantly outperforming related methods as well as a GPT-3 neural model for the detection of *values*, *units* and *changes* but also introduces the novel task of *concept* detection. Furthermore, we present the first benchmark dataset for the comprehensive evaluation of quantity extraction and make our code and data available to the community. We are currently extending the extractor by improving the quality of edge cases and looking at the compatibility of our rule set to other application domains, e.g., medical text.

## 6 Limitations

Despite an extensive effort to account for most common cases, CQE is still mainly a rule-based approach, requiring manual feature engineering and rule-writing for unseen cases. This issue is more prominent in the case of *concept* extraction, where the order in which we apply the rules has a direct impact on correct extractions. If the rule with higher priority finds a candidate, the rules further down the list are ignored. Although for humans identifying the correct rule to use is easy by considering context and sentence formulation, such delicate difference in language is not easily captured in rule-based systems. Moreover, CQE relies heavily on correct dependency parsing and POS tagging, and any error on the initial extraction propagates through the entire system. Consequently, even changes in the versions of the SpaCy model used for dependency parsing and POS tagging can produce slightly varying results.

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

# A  Appendix

## A.1  Value, Unit and Change Detection Rule

In this section, we provide two additional examples for *value*, *unit*, and *change* detection and describe the logic behind a few other rules.

**Example 2:** "The Meged field has produced in the past about 1 million barrels of oil, but its last well was capped due to technical problems that have not been resolved."

- NUM_NUM detects the compound number of 1 *million*, where 1, a number, is the child of *million*, a noun, in the dependency tree.

- QUANTMOD_DIRECT_NUM detects the relation between the adjective "about" to the value 1, which is later identified as the change.

- NOUN_NUM_ADP_RIGHT_NOUN finds a noun or proper noun that has a number as a child in the dependency tree. If there are prepositions in the children of the noun, they are also considered part of the unit. In this case, [*million, barrels, of, oil*] are detected using this rule.

The naming of the rules is preserved in the repository. From the combination of all rules, the candidate tokens [1, *million*] for *value*, [*barrels, of, oil*] for *unit* and [*about*] for change are extracted.

**Example 3:** "They have a $3500 a month mortgage and two kids in private school."

- NUM_SYMBOL matches a symbol followed by a number. In this case, $3500 is detected.

- NOUN_NUM_QUANT finds a number with a noun or an adverb as its head in the dependency tree. Here, we have [*mortgage, 3500, $, month*].

- UNIT_FRAC_2 finds compound units with "per", "a" or "an" in between, e.g., [3500, *month, a*]

- NOUN_NUM detects a noun that has a number as a child, e.g., [*kids, two*].

The mentioned rules contribute to extraction of two candidate quantities: [[$, 3500, *a, month, mortgage*], [*two, kids*]].

## A.2  GPT-3 and Few-shot Learning

To tag sentences using GPT-3, we use the few-shot learning paradigm by prompting the model to tag quantities and units in the text, given 10 distinct examples. GPT-3 is mainly advertised as a task-agnostic, few-shot learner, and we have not performed extensive fine-tuning. With the 10 examples, we aim to account for a variety of outputs, e.g., compound units, when no quantity is present, noun-based units, and prefixes for scaling the magnitude of a value. Our full prompt is as follows, where the quantities are output in a numbered list, with an order of *change*, *value*, unit surface form, *unit*, *concept*. The unit surface form is used in post-processing if GPT-3 is not able to normalize the unit.

```
Tag quantities and units in the texts:

Sentence: Woot is selling refurbished,
unlocked iPhone XR phones with 64GB of
storage for about $330.
Answer:
1. =, 1.64, GB, gigabyte, storage
2. ~, 330, $, dollar, iPhone XR phones

Sentence: The chain operates more than 600
supermarkets and less than 800 convenience
stores.
Answer:
1. >, 600, supermarkets, supermarkets,
chain
2. <, 800, convenience stores, convenience
stores, chain

Sentence: The spacecraft, which is about
the size of a school bus, flew into
Dimorphos at a speed of about 4.1 miles
per second, that's roughly 14,760
miles per hour (23,760 kilometers per
hour).
Answer:
1. ~, 4.1,  miles per second, mile per
second, spacecraft
2. ~, 14760, miles per hour, mile per hour,
spacecraft
3. ~, 23760, kilometers per hour, kilometer
per hour, spacecraft

Sentence: And overnight dogecoin fell from
0.317 to 0.308, a 2.8 percent drop.
Answer:
```

```
1. =, 1.0.317-0.308, -, -, dogecoin
2. =, 2.8, percent, percentage, dogecoin
Sentence: This is about minus 387
Fahrenheit (minus 233 Celsius).
Answer:
1. ~, -387, Fahrenheit, Fahrenheit, -
2. ~, -233, Celsius, Celsius, -

Sentence: WhatsApp more than 2 billion
users send  fewer than 100bn messages a day.
Answer:
1. >, 2000000000, users, users, WhatsApp
2. <, 100000000000, messages, messages,
users

Sentence: This includes colors between red
and blue - wavelengths ranging between 390
and 700 nm.
Answer:
1. =, 390-700, nm, nanometer, wavelengths

Sentence: You don't have a two-year
bachelor's degree or a six to eight-year
 phd degree.
Answer:
1. =, 2, year, year, bachelors degree
2. =, 6-8, year, year, phd degree

Sentence: The price CO2 and fuel
consumption are not clear.
Answer:
No quantities or units

Sentence:{sentence}
Answer:
```

{sentence} is replaced with the query sentence to be tagged. Nevertheless, the output of GPT-3 is not consistent and requires extreme post-processing. The post-processing includes cleaning the predicted values to only include numbers, normalization of the units even if the unit is miss-spelled, e.g., "celsiu" instead of "celsius", "ppb" to "parts-per-billion", or "€" to "euro".

## A.3 Ambiguous Surface Forms

In our unit dictionary, we encountered 18 ambiguous surface forms with different normalized units and collected at least 100 samples for each. This list is not comprehensive and in different scientific domains, more ambiguous cases might occur. The

Table 7: Ambiguous surface forms, *units* associated with them and the number of samples in the training set for each surface form and unit pair.

| Surface | Units | # samples |
|---------|-------|-----------|
| c | cent, celsius | 144 |
| ¥ | chinese yuan, japanese yen | 100 |
| kn | croatian kuna, knot | 116 |
| p | point, penny | 149 |
| R | south african rand, roentgen | 100 |
| b | barn, bit | 127 |
| ' | foot, minute | 104 |
| ′ | foot, minute | 104 |
| " | inch, second | 112 |
| " | inch, second | 112 |
| C | celsius, coulomb | 116 |
| F | fahrenheit, farad | 100 |
| kt | kiloton, knot | 100 |
| B | byte, bel | 107 |
| P | poise, pixel | 102 |
| dram | armenian dram, dram | 180 |
| pound | pound sterling, pound-mass | 131 |
| a | acre, year | 113 |

number of samples per surface form and associated units for each surface form are shown in Table 7.

## A.4 Disambiguation Prompts

To generate the dataset for disambiguation, we experimented with multiple prompts, using ChatGPT. The aim was to create training/test data in JSON-format, where the sentences are not duplicates or too simple. For this purpose, two sentences were formulated (one for each unit, in each surface form) and are used as input examples of different contexts. The prompt explicitly asks for JSON format output and 20 samples, due to the sequence length limitation of ChatGPT. The final prompt is as follows, where UNIT1 and UNIT1 are replaced with different units with the shared surface and "SURFACE_FORM" denotes the ambiguous surface form:

```
Create a training set of 20 samples, for
"UNIT1" and "UNIT2", where in the text the
surface form of the unit is always
"SURFACE_FORM", but the unit is different.
Output in JSON format as follows:

{"text":"Sentence 1", "unit": "UNIT1" },
{"text":"Sentence 2 ", "unit": "UNIT2" }}
```

The test dataset is created in the same manner. For certain units, multiple generations were required to get more complex sentences. In such cases, we

Table 8: Error analysis of different extraction systems.

| Mistake | Systems |
| --- | --- |
| Trouble detecting temperature types, e.g., celsius and fahrenheit are both denoted as degree. | Q3, IllQ, GPT-3, Grbd |
| Dollar types are not identified, e.g.,"hong kong dollar" and "new zealand dollar" → dollar. | Q3, IllQ, Grbd |
| Unit normalization does not work for the majority of the times. | IllQ, GPT-3, Grbd |
| Bias towards predicting = for changes. | IllQ, GPT-3 |
| Cryptocurrencies and rare ones are not recognized, e.g. Bitcoin or Markka. | Q3 |
| Sports units are not recognized, e.g., ppg, rpg, apg. | Q3 |
| Temporal values are mistaken as quantities, e.g., 2 pm. | Txt-R, GPT-3, Grdb |
| Compound units are rarely found, e.g., kph. | Txt-R, GPT-3 |
| Units in short sentences are not recognized, e.g., rmb 10 usd 20. | CQE |
| Problematic distinction between "year" and "year of age". | CQE, Q3, GPT-3 |
| Units are confused with concepts, e.g., building rate of $80 per sq m → "sq m" as a concept. | GPT-3 |
| Low recall due to limited quantity types. | Txt-R, Grbd |
| Detection of concepts where none exist. | CQE |
| Problem with correct standardization of values. | Grdb |
| Multi word compound units and most percentages are ignored. | Grdb |
| Unable to correctly distinguish different temperature units. | Grdb |

specifically asked for sentences that do not start with "the" and are more complex. After each generation, all examples were checked by the authors of the paper. Faulty samples with wrong units were removed. In some cases, surface forms were manually altered to match the specifications of the task.

### A.5 Disambiguation per Class

A detailed evaluation of the disambiguation dataset is shown in Table 9, where precision, recall, and F1-score are computed separately for each class. For each surface form, 10 examples are present in the test dataset. We noticed that distinguishing between "Japanese yen" and "Chinese yuan" is partially difficult for the BERT-based classifier since both of them are currencies and used in similar contexts. Another difficult distinction is between "penny" and "point", since monetary values and the stock market point unit are used in similar contexts. In comparison, in Q3 certain units are almost never predicted, hence the multiple zeros in the evaluation results.

### A.6 Error Analysis on NewsQuant and Txt-R Datasets

We analyzed the incorrect detection for all the models and the common mistakes. Except for the points discussed in the main part of the paper, Table 8 provides an overview of the remaining common mistakes and systems associated with them.

### A.7 Output Post-processing

The goal of the post-processing is to achieve a unified representation, close to the benchmark dataset.

Since baseline models have various definitions of quantities and different unit dictionaries, it is expected that their output representation varies although in many cases they refer to the same quantity. This is most prominent for the units, as different unit dictionaries or lack of unit normalization results in multiple surface forms for a single unit. For example, the unit "celsius" in the benchmark dataset is detected as "degree celsius" by Q3, "c" by Txt-R, "c", "° celsiu" and"celsiu" by IllQ, "celsiu", "degrees celsiu" and "degree celsiu" by GPT-3, "degc" and "degC" by Grbd. As demonstrated in the example, GPT-3, Grbd, and IllQ not only have a single different name for the unit celsius but multiple surface forms. Although one can argue that multiple representations for the is pointing to a lack of normalization altogether, we chose to map these variations to a single unified format, such that they are all pointing to the same unit. The only exception is for currencies in IllQ, since their work explicitly claims that unit normalization is performed for currencies. For the entire pipeline and post-processing steps, refer to the evaluation repository.

Table 9: Precision, recall and F1-score for *unit* disambiguation per class.

| Class | CQE | | | Q3 | | |
|---|---|---|---|---|---|---|
| | P | R | F1 | P | R | F1 |
| knot | 100 | 100 | 100 | 16.67 | 100 | 28.57 |
| roentgen | 100 | 100 | 100 | 44.44 | 80 | 57.14 |
| barn | 100 | 80 | 88.89 | 100 | 80 | 88.89 |
| Japanese yen | 60 | 100 | 75 | 100 | 100 | 100 |
| inch | 83.33 | 100 | 90.91 | 77.78 | 70 | 0.7368 |
| Armenian dram | 100 | 60 | 75 | 0 | 0 | 0 |
| Chinese yuan | 0 | 0 | 0 | 100 | 100 | 100 |
| byte | 100 | 100 | 100 | 57.14 | 80 | 66.67 |
| cent | 83.33 | 100 | 90.91 | 0 | 0 | 0 |
| Croatian kuna | 100 | 100 | 100 | 0 | 0 | 0 |
| year | 100 | 100 | 100 | 0 | 0 | 0 |
| poise | 100 | 100 | 100 | 100 | 100 | 100 |
| south african rand | 100 | 100 | 100 | 100 | 60 | 75 |
| minute | 80 | 80 | 80 | 60 | 90 | 72 |
| bit | 83.33 | 100 | 90.91 | 50 | 40 | 44.44 |
| bel | 80 | 100 | 88.89 | 100 | 100 | 100 |
| kiloton | 100 | 100 | 100 | 100 | 100 | 100 |
| second | 100 | 80 | 88.89 | 80 | 40 | 53.33 |
| coulomb | 100 | 100 | 100 | 100 | 100 | 100 |
| dram | 71.43 | 100 | 83.33 | 0 | 0 | 0 |
| point | 55.56 | 100 | 71.43 | 0 | 0 | 0 |
| fahrenheit | 100 | 100 | 100 | 100 | 100 | 100 |
| celsius | 100 | 90 | 94.74 | 0 | 0 | 0 |
| pixel | 100 | 100 | 100 | 80 | 80 | 80 |
| pound-mass | 100 | 100 | 100 | 83.33 | 100 | 90.91 |
| pound sterling | 100 | 100 | 100 | 100 | 80 | 88.89 |
| foot | 80 | 80 | 80 | 100 | 50 | 66.67 |
| penny | 100 | 20 | 33.33 | 0 | 0 | 0 |
| farad | 100 | 80 | 88.89 | 80 | 80 | 80 |
| acre | 100 | 100 | 100 | 0 | 0 | 0 |