# OpenReview forum: "CQE: A Comprehensive Quantity Extractor"
_EMNLP/2023/Conference — EMNLP 2023 Main_

### Official Review · Reviewer_vmbW · 2023-07-24

**Soundness:** 3

**Excitement:**

3: Ambivalent: It has merits (e.g., it reports state-of-the-art results, the idea is nice), but there are key weaknesses (e.g., it describes incremental work), and it can significantly benefit from another round of revision. However, I won't object to accepting it if my co-reviewers champion it.

**Paper Topic And Main Contributions:**

The paper presents a new method called CQE for the extraction of mentions of quantity, unit, and concept in textual documents. CQE consists of a set of 61 syntax-based rules and a BERT-based classifier for unit disambiguation. The classifier is trained with examples automatically generated using ChatGPT.
The paper also presents NewsQuant, which is a new corpus manually annotated with quantity, unit, and concept mentions where quantity and units are normalized.
CQE is compared to three other quantity extractors and GPT3. The experiments show that CQE outperforms them on NewsQuant.

**Questions For The Authors:**

A.	Please define “helper tokens”.

B.	Could you explain why the method computes the similarity between subclauses by comparing sequences instead of parsed trees? (section 3.2.3)

C.	Line 366-367: “the conversion between metric units is not supported”. Is the conversion of other units, such as volumes (ml, l) supported?

D.	Why has Grobid-quantities method cited in the Related Work section (line 87) not been compared?

E.	Are units expressed by Greek letters handled by CQE?

**Reasons To Accept:**

The extraction of quantities, units, and the entity measured is a difficult problem that has many applications in science, technics, and the general domain. There are relatively few methods and corpora available. This paper describes a new work for the English language that will be made available on GitHub.

The method is compared to four state-of-the-art methods that it outperforms. The paper is globally clear and convincing. The experiments are detailed. The result analysis section and the A6 appendix provide relevant hypotheses and examples.

**Reasons To Reject:**

The notion of ‘concept’ is insufficiently described and formalized. Concepts are defined as “either properties being measured or entities that the value is referring to or is acting upon”, which is vague. The CQE method is designed “to capture as many concepts as possible.” This has consequences for the manual annotation of the NewsQuant corpus where the inter-annotator agreement is low, and for the comparison of CQE and GPT3 (Table 5), which is based on a questionable reference.

More generally, a more formal definition would make the evaluation of the method and, furthermore, its comparison with other methods more robust. It would also broaden the scope of application. As a source of inspiration, the authors could look at SSN and SOSA that distinguish between the measured property and the entity (or part of entity) observed and spatiotemporal conditions.

The same remark applies to ‘change’. IllQ achieves lower performances for Value+change extraction. How much could it be due to a different understanding of what ‘change’ is? The definition of IllQ for the ‘change’ entity would be needed here.

It is standard to provide the annotators with written guidelines, along with the corpus distribution. It would improve the annotation quality and reusability.

The reader understands that the outputs of the methods are not directly comparable, and some post-processing is applied to unify them that add information to the direct output. This is not a correct procedure, whatever is the post-processing is. The comparison should be done on the minimal common subset instead of “improving” the existing methods to mimic the CQE results.

More precisely, the comparison of units extracted by IllQ and Q3 and by CQE as described in section 4.4 Implementation seems unfair and must be clarified. The reader understands that the outputs of the three methods are not directly comparable, and some post-processing is applied to normalize and unify them. The paper must provide the performance score of the extraction of units of the three methods without alteration for a fair comparison. The post-processing needed for comparing the normalized extracted units must be detailed and show that no bias is introduced. The mention “Unit normalization does not work for the majority of the times” for IllQ and GPT3 should be replaced by “Unit normalization not available” to be consistent with Table 1 (line 10).

Another postprocessing is added to R-Txt model (Line 478-479) that adds extra annotations for all types of quantities. This makes R-Txt and CQE methods not comparable. The evaluation should have been done for the only quantities that R-Txt can predict.

**Reproducibility:**

4: Could mostly reproduce the results, but there may be some variation because of sample variance or minor variations in their interpretation of the protocol or method.

**Reviewer Confidence:**

4: Quite sure. I tried to check the important points carefully. It's unlikely, though conceivable, that I missed something that should affect my ratings.

**Typos Grammar Style And Presentation Improvements:**

About the formalization of units and linked concepts, these three papers could be useful to look at.

Rijgersberg, H., Van Assem, M., & Top, J. (2013). Ontology of units of measure and related concepts. Semantic Web, 4(1), 3-13.
Janowicz, K., Haller, A., Cox, S. J., Le Phuoc, D., & Lefrançois, M. (2019). SOSA: A lightweight ontology for sensors, observations, samples, and actuators. Journal of Web Semantics, 56, 1-10.

Compton, M., Barnaghi, P., Bermudez, L., Garcia-Castro, R., Corcho, O., Cox, S., ... & Taylor, K. (2012). The SSN ontology of the W3C semantic sensor network incubator group. Journal of Web Semantics, 17, 25-32.

About the extraction of units, measures, and entities, the reference [Berrahou et al., 2017] may interesting to look at.

Berrahou, S. L., Buche, P., Dibie, J., & Roche, M. (2017). Xart: Discovery of correlated arguments of n-ary relations in text. Expert Systems with Applications, 73, 115-124.

---

> ### Author Rebuttal · Authors · 2023-08-25
>
> Thank you very much for the detailed comments, clarifying questions, and pointers to useful related work. The related work section will be adjusted accordingly.
>
> Before answering the individual questions, first some general statements:
>
> We were somewhat surprised by the low reproducibility score since both the code and the dataset were submitted alongside the paper (please refer to the zip file in supplementary material on open review platform). The other reviewers were quite happy with the access and reproducibility.
>
> We understand the concern you raised regarding the comparison of models with different outputs. However, claiming that a "comparison should be done on the minimal common subset" is somewhat misleading. We argue that one should not design a benchmark to match the minimum requirement of the systems that came before, but rather design a comprehensive benchmark, encompassing essential information about a quantity even if previous systems are not able to extract all of it. To this end, the aim of post-processing is not to mimic the CQE results but rather to extract information in the standardized way of the benchmark.  Without the post-processing, models that do not perform unit normalization could have not been compared at all.
>
> This also brings us to the comparison against the  R-Txt model. In the same way that lexical based systems like BM25 are still used as baseline against large language models with semantic understanding, one should not degrade and limit a model to a lower standard because another system cannot compete. The aim of this paper, beyond introducing a novel system, is to point out the lack of a benchmark and consolidate definitions and evaluation systems for quantity extraction, and the R-Txt model with such vast limitations is a good demonstration of the importance of this task.
>
> We also appreciate your feedback regarding the clarification of the notation and agree that we should elaborate more on it.
> In simple terms, a quantity mentioned in the text is either measuring a property of a phenomenon, e.g., ''height of the Eiffel Tower", in which case the phenomenon and the property are the concepts, or an action has been made involving a quantity, e.g, "Google hired 100 people", in which case the actor is what the quantity is referring to.
> Our definition of change aligns directly with IllQ, which, to quote directly from their paper, says “A change specifies how the parameter is changing, e.g., increasing.”
>
> In the following, we will address the questions raised:
> A. CQE relies heavily on POS tagging and dependency parsing performed by the SpaCy library, where certain patterns in text result in incorrect extractions. For example, with negative values such as  -100, the token "-" is treated as a punctuation separate from the value. To this end, during pre-processing, we change "-" to minus for a correct extraction.
>
> B. Investigating the parsing tree is also a viable way of looking at subclauses. However, looking at part-of-speech tags provide a fast and effective method in our case. We did not investigate alternatives.
>
> C. While metric units can be converted based on conversion factors, conversions are not performed and are not the focus of this study. The reason for this decision is two-fold: 1) Converting all units to a base unit would remove the important unit granularities that are specific to certain domains. 2) Inter-unit conversion is achievable by libraries or a dictionary of scaling factors and is not a novel contribution of the system.
>
> D. Very good point and one we should have mentioned in the paper. Upon the original submission of the paper, we were not able to run the code from the Gorbid quantities. However, the repository was updated two months ago and we now performed additional evaluations, which we will include in the paper. We include the results in the following:
>
> NewsQuant:
>
> 	P	R	F1	model	type
> 	58.8	53.1	55.8	Gorbid	value
> 	37.4	33.7	35.5	Gorbid	value+unit
>
> R-Txt-age:
>
> 	P	R	F1	model	type
> 	60.0	27.3	37.5	Gorbid	value
> 	40.0	18.2	25.0	Gorbid	value+unit
>
> R-Txt-temperature:
>
> 	P	R	F1	model	type
> 	84.0	61.8	71.2	Gorbid	value
> 	32.0	23.5	27.1	Gorbid	value+unit
>
> R-Txt-dimension:
>
>
> 	P	R	F1	model	type
> 	75.8	59.5	66.7	Gorbid	value
> 	44.2	34.7	38.9	Gorbid	value+unit
>
> R-Txt-currency:
>
> 	P	R	F1	model	type
> 	46.6	35.3	40.2	Gorbid	value
> 	24.9	18.8	21.4	Gorbid	value+unit
>
> Comparing the result with the tables in the paper shows that Gorbid quantities performs on par with other baselines. The model has a particular problem with the detection of currencies and percentages in the text. The unit and value normalization has different meanings in their system, the authors do not normalize the name of the units to a unified representation, they rather convert metric units to a base unit (as indicated in the answer to C) and scale the values accordingly.
>
> E. Yes, they are.

---

### Official Review · Reviewer_H5PX · 2023-08-05

**Soundness:** 4

**Excitement:**

4: Strong: This paper deepens the understanding of some phenomenon or lowers the barriers to an existing research direction.

**Missing References:**

As far as I could tell, the presented references are enough.


**Paper Topic And Main Contributions:**

This paper describes an interesting solution for a seemly simple problem with a dedicated approach to extract quantities, units, modifiers, and concepts from textual sources.


**Questions For The Authors:**

No specific question to ask.

**Reasons To Accept:**

This work is of a refreshing return to ingenious solutions to problems seemly simple, but that can elude state of the art solutions. The paper is well written and easy to follow. The developed work is sound and the originality resides in narrowing the application to a single problem and solving it with a high precision and recall compared to baselines.


**Reasons To Reject:**

The options and decisions taken could have been more explored and analyzed in greater detail.


**Reproducibility:**

4: Could mostly reproduce the results, but there may be some variation because of sample variance or minor variations in their interpretation of the protocol or method.

**Reviewer Confidence:**

3: Pretty sure, but there's a chance I missed something. Although I have a good feel for this area in general, I did not carefully check the paper's details, e.g., the math, experimental design, or novelty.

**Typos Grammar Style And Presentation Improvements:**

As far as I could tell, there were no typos or grammar issues.

---

### Official Review · Reviewer_5wNU · 2023-08-05

**Soundness:** 5

**Excitement:**

5: Transformative: This paper is likely to change its subfield or computational linguistics broadly. It should be considered for a best paper award. This paper changes the current understanding of some phenomenon, shows a widely held practice to be erroneous in someway, enables a promising direction of research for a (broad or narrow) topic, or creates an exciting new technique.

**Paper Topic And Main Contributions:**

This paper presents a novel framework for quantity extraction from unstructured text, utilising dependency parsing and a dictionary of units. The key contributions include a novel methodology for detecting concepts associated with identified quantities and introducing a new benchmark dataset for evaluating quantity extraction methods.

**Reasons To Accept:**

The paper lays a strong mathematical foundation for the proposed methodology, ensuring its reproducibility. The experimental design and results are well-documented, enabling a clear understanding to the reader. Moreover, the introduction of a new benchmark dataset is a significant contribution to the field.

**Reasons To Reject:**

While the paper presents a sound framework, including a visual representation of the proposed methodology would have been beneficial.

**Reproducibility:**

4: Could mostly reproduce the results, but there may be some variation because of sample variance or minor variations in their interpretation of the protocol or method.

**Reviewer Confidence:**

4: Quite sure. I tried to check the important points carefully. It's unlikely, though conceivable, that I missed something that should affect my ratings.

---

### Meta-Review · Area_Chair_1jC6 · 2023-09-24

**Recommendation:** 5

**Metareview:**

This paper introduces a novel framework for extracting quantities from unstructured text using dependency parsing and a unit dictionary. It also presents a new benchmark dataset for evaluating quantity extraction methods. Its soundness and excitement are generally well-supported by reviewers because of the simplicity and insight provided by this work. Two major areas of improvements are brought by one of our reviewers. (1) The definitions of "concept" and "change" are important for researchers and annotators, which are not articulated very clearly in the manuscript; (2) the use of post-processing to convert output from existing works may lead to unfair comparisons so it's suggested that the authors include a thorough explanation of the post-processing steps.

---

### Decision · Program_Chairs · 2023-10-07

**Decision:**

Accept-Main

**Comment:**

This paper introduces a novel framework for extracting quantities from unstructured text using dependency parsing and a unit dictionary. It also presents a new benchmark dataset for evaluating quantity extraction methods. Its soundness and excitement are generally well-supported by reviewers because of the simplicity and insight provided by this work. Two major areas of improvements are brought by one of our reviewers. (1) The definitions of "concept" and "change" are important for researchers and annotators, which are not articulated very clearly in the manuscript; (2) the use of post-processing to convert output from existing works may lead to unfair comparisons so it's suggested that the authors include a thorough explanation of the post-processing steps.